# Switching nanoprecipitates to resist hydrogen embrittlement in high-strength aluminum alloys

Yafei Wang [1,2,7] ✉, Bhupendra Sharma [1,7] ✉, Yuantao Xu [1,3] ✉, Kazuyuki Shimizu [4], Hiro Fujihara[1], Kyosuke Hirayama[5], Akihisa Takeuchi [6], Masayuki Uesugi [6], Guangxu Cheng[2] & Hiroyuki Toda[1]

Hydrogen drastically embrittles high-strength aluminum alloys, which impedes efforts to develop ultrastrong components in the aerospace and transportation industries. Understanding and utilizing the interaction of hydrogen with core strengthening elements in aluminum alloys, particularly nanoprecipitates, are critical to break this bottleneck. Herein, we show that hydrogen embrittlement of aluminum alloys can be largely suppressed by switching nanoprecipitates from the η phase to the T phase without changing the overall chemical composition. The T phase strongly traps hydrogen and resists hydrogen-assisted crack growth, with a more than 60% reduction in the areal fractions of cracks. The T phase-induced reduction in the concentration of hydrogen at defects and interfaces, which facilitates crack growth, primarily contributes to the suppressed hydrogen embrittlement. Transforming precipitates into strong hydrogen traps is proven to be a potential mitigation strategy for hydrogen embrittlement in aluminum alloys.

Aluminum alloys with high and ultrahigh strengths are highly attractive for weight-sensitive structures such as aircraft, bullet trains, and automobiles. However, unlike the significantly elevated strength of steels up to ~1.5 GPa or even higher levels[1], the development of ultrastrong aluminum alloys has almost stagnated in the past decades, largely due to the strength-hydrogen embrittlement (HE) conflict: with increasing strength, hydrogen drastically reduces the ability of aluminum alloys to sustain plastic deformation and cyclic load[2]. Such an effect arises from the interaction of hydrogen atoms with various micro- or nanoscale structures, i.e., so-called hydrogen trap sites, including defects in lattices, e.g., dislocations[3,4] and vacancies[5,6], grain boundaries[7,8], interfaces of micron-scale particles[9,10] and interfaces of strengthening nanoprecipitates[11]. Through these pathways, a macroscopically premature fracture can be triggered,

accompanied by internal quasi-cleavage and/or intergranular cracks (IGCs)[12,13]. The HE phenomenon was first reported for iron[14] and evidenced for different metallic materials[15,16], particularly for high-strength aluminum alloys[17] due to the easy generation and ingress of hydrogen via aluminum-water reactions.

Despite the persistent debate on HE mechanisms, the search for mitigation strategies is crucially important to realize ultrahigh strengthening of aluminum alloys and prevent disastrous failures. A viable route for HE suppression is adding new chemical elements to the bulk material to develop deep trap sites that can attract a large number of hydrogen atoms via strong atomic bonds, so that the percentage of occupied hydrogen sites (hydrogen occupancy) at various defects and interfaces can be reduced based on the thermal equilibrium relationships among hydrogen trap sites[18]. With the advance of simulation techniques, the trapping capacity of candidate hydrogen

[1]Department of Mechanical Engineering, Kyushu University, Fukuoka 819-0395, Japan. [2]School of Chemical Engineering and Technology, Xi'an Jiaotong University, Xi'an 710049, China. [3]Shanghai Key Laboratory of Materials Laser Processing and Modification, Shanghai Jiao Tong University, Shanghai 200240, China. [4]Department of Physical Science and Materials Engineering, Iwate University, Iwate 020-8551, Japan. [5]Department of Materials Science and Engineering, Kyoto University, Kyoto 606-8501, Japan. [6]Japan Synchrotron Radiation Research Institute, Hyogo 679-5198, Japan. [7]These authors contributed equally: Yafei Wang, Bhupendra Sharma. ✉e-mail: yafeiwang90@outlook.com; sharma.bhupendra.464@m.kyushu-u.ac.jp; xu.yuantao.158@m.kyushu-u.ac.jp

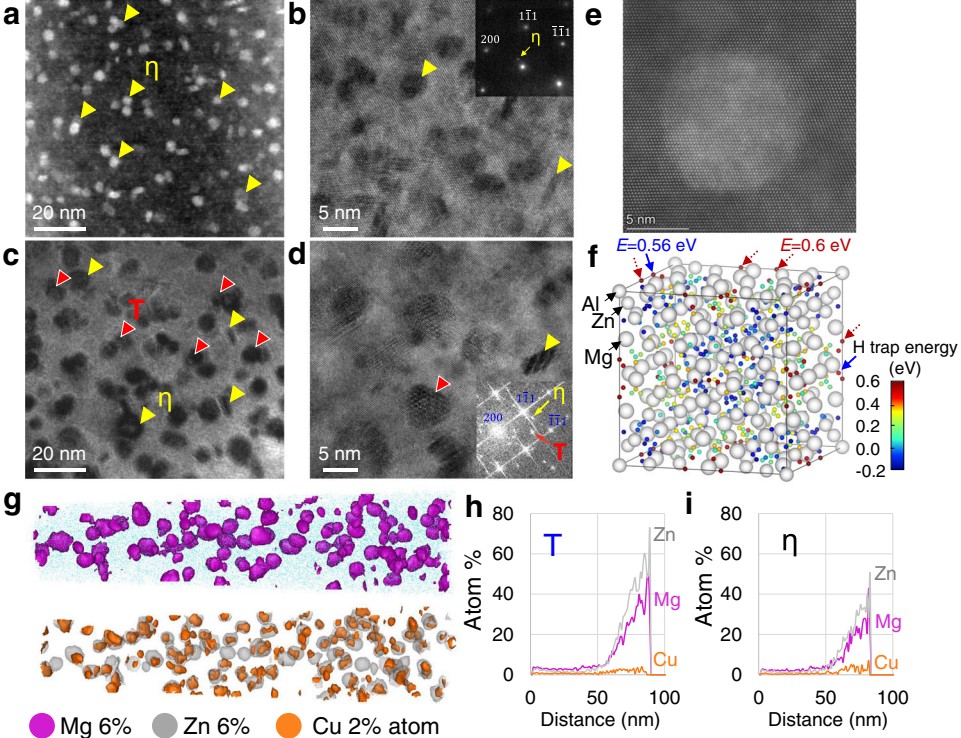

**Fig. 1 | Morphologies of precipitates and hydrogen trapping at T phase in Al-Mg-Zn-Cu alloys. a** TEM image and **b** diffraction pattern of η phase in the LT material; **c** TEM image and **d** fast Fourier transform of T phase in the HT material. **e** magnified HAADF-STEM image of the T phase. All images were taken along the [110]$_{Al}$ zone axis. **f** distribution of hydrogen trapping energies in the interior of the T phase with a maximum energy of 0.6 eV predicted by first-principles calculations. **g** element maps for Mg, Zn, and Cu with 6, 6, and 2% iso-surfaces obtained by APT. **h**, **i** cross-sectional atom percentages of Zn, Mg, and Cu in T and η precipitates.

traps can be theoretically explored, and their practical effects can be further examined in experiments. Following this route, successful examples have been reported for steels, including NbC precipitates with their strong hydrogen trapping effect predicted by first-principles simulations[19] and experimentally verified by atom probe tomography (APT)[20]. Similarly, vanadium was found to be beneficial for resisting HE in steels[21], which was rationalized by the hydrogen trapping at vanadium carbides seen in simulations[22] and experiments[23]. Unfortunately, no ideal trap sites have been reported and confirmed for aluminum alloys until now.

One step forward in this direction is the newly reported beneficial effects of intermetallic compound particles in trapping hydrogen and reducing quasi-cleavage cracks in high-strength aluminum alloys[24,25]. However, most coarse particles are intrinsically brittle[26], and their dynamic hydrogen trapping ability is weakened by their large size since full trapping of hydrogen atoms within μm-sized particles in a short period of time is difficult; instead, hydrogen can rapidly diffuse to extremely small-sized (down to several layers of atoms) stressed regions and readily facilitate crack growth[27]. This necessitates the search for effective nanosized structures for hydrogen trapping and HE mitigation in aluminum alloys. Typically, high-strength aluminum alloys are developed by age-hardening methods, which involve heat treatment at a specific temperature to induce a transformation from soluble elements in a supersaturated solid solution to insoluble phases. These insoluble nanoparticles, so-called precipitates, impede the movement of lattice defects and lead to high strength of aluminum alloys. It is thereby believed that the basis of HE mitigation methods must lie in the modification of hydrogen-precipitate interactions to solve the strength-HE conflict.

Here, we show that nanosized age-hardening precipitates, widely available and serving as core-strengthening elements in high-strength aluminum alloys, can be switched into strong hydrogen trap sites and contribute to elevated HE resistance. Bearing in mind the detrimental effects of η precipitates (and their numerous variants, here unified as η-MgZn$_2$) in HE due to the risk of interfacial debonding[11], we show the strong and beneficial hydrogen trapping effect of T (Al$_2$Mg$_3$Zn$_3$) precipitates. We further investigate the effectiveness of these precipitates in the control of HE and related mechanisms by advanced characterization techniques, taking a typical Al-Mg-Zn-Cu aluminum alloy as a model material.

## Results

The quaternary 7XXX Al alloys with a chemical composition of Al-5.6Zn-2.5Mg-1.6Cu (wt%) were prepared such that the η phase was partially switched to the T phase through modification of the aging parameters (details shown in Methods and Supplementary Fig. 1) without changing its overall chemical composition and texture. We hypothesized a partial transformation from η to T when the aging temperature was elevated from low temperature (LT) to high temperature (HT), which was confirmed by transmission electron microscopy (TEM) and selected area electron diffraction patterns, indicating fully η phase in the LT material (Fig. 1a, b) and a considerable amount of T phase in the HT material (Fig. 1c, d and Supplementary Fig. 2). The typical high-resolution morphology of a T precipitate is shown in Fig. 1e, exhibiting the near-spherical shape in contrast with the elongated η phase. More high-angle annular dark field-scanning transmission electron microscopy (HAADF-STEM) images and fast Fourier transform analyses are included in Supplementary Fig. 3, for the confirmation of the T phase. The first-principles simulation shown in Fig. 1f indicates excellent hydrogen trapping capacity in the interior of T precipitates with a maximum binding energy of 0.6 eV. The APT results in Fig. 1g to i provide compositional evidence of the T phase (with a maximum Zn + Mg concentration close to 80 at.%) in HT material[28]. The increase in aging temperature did not significantly alter the precipitate size (with an average diameter lower than 7 nm in both), which

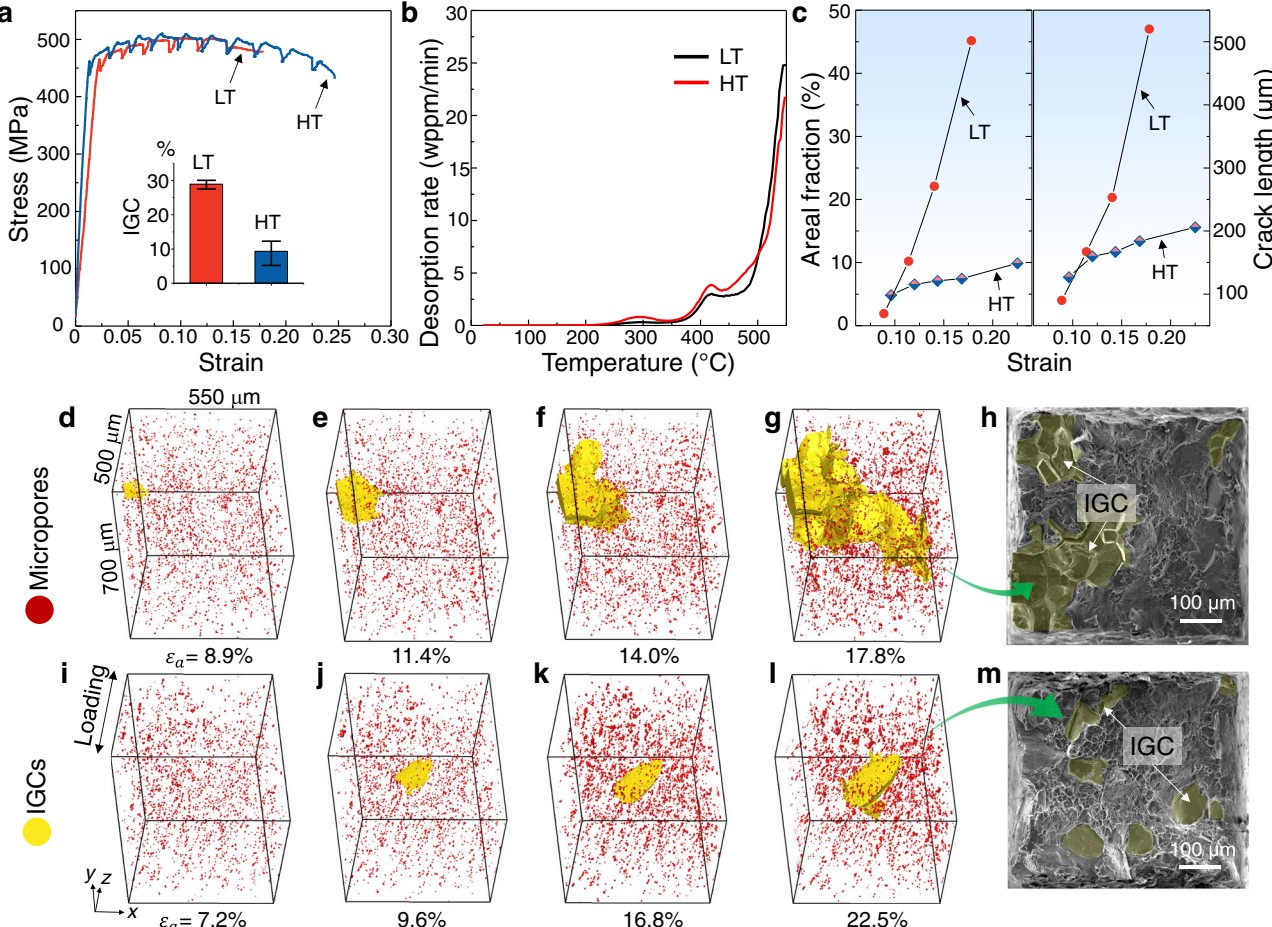

**Fig. 2 | Mechanical properties and crack growth behavior. a** Stress−strain curves for LT and HT materials, with the inset figure showing the average areal fractions of IGCs on the fracture surfaces. Error bars are determined from repeat tensile tests. **b** Thermal desorption curves of hydrogen-charged LT and HT materials. **c** Dependence of areal fraction and crack length of IGCs on the applied strain for LT and HT materials. **d**−**g** 3D renderings of the IGCs in the LT material at an applied strain of $\varepsilon_a$ = 8.9, 11.4, 14.0, and 17.8%, respectively. **h** SEM morphologies of the fracture surface for the LT material. **i**−**l** 3D renderings of the IGCs in the HT material at an applied strain of $\varepsilon_a$ = 7.2, 9.6, 16.8, and 22.5%, respectively. **m** SEM morphologies of the fracture surface for the HT material.

enables evaluation of the HE sensitivities of these two materials with similar tensile strengths and precipitate coherency[29,30], although some of the coarse η precipitates in HT material are semi-coherent (Supplementary Fig. 4).

In situ tensile tests under synchrotron X-ray tomography (see Supplementary Fig. 5 for the experimental setup) indicated significantly improved ductility due to the change in nanoprecipitates (Fig. 2a). At the same hydrogen content level (Fig. 2b), the presence of the T phase resulted in a 38% increase in the fracture strain and a more than 60% reduction in the areal fractions of IGCs on the fracture surface. 4D observations directly proved the significantly reduced growth rate of the main crack in the presence of the T phase in terms of both areal fraction and crack length in 2D projections (Fig. 2c). In the last loading step, the areal fraction of the main crack decreased from 45% to less than 10%, and the crack length decreased from more than 500 to 200 µm. In contrast to the hydrogen-induced fast grain boundary (GB) separation in the LT material (Fig. 2d, g), the main crack in the T phase-rich material remained almost stagnant with increasing applied strain until the final fracture (Fig. 2I, l). Scanning electron microscopy (SEM) images of fracture surfaces obtained from repeat tensile tests (Fig. 2h, m and Supplementary Fig. 6) demonstrated the T phase-induced transition from large-area IGCs to small-sized separate cracks. According to the decohesion mechanism[31], hydrogen reduces the energy required to separate various interfaces (cohesive energy), including GBs, and hydrogen coverage at GBs controls the growth

behavior of IGCs, such as the crack propagation velocity[32]. The intense stress field in the vicinity of a GB ahead of the crack tip, after the initiation of an IGC, attracts more hydrogen atoms toward it[33], and the crack growth speed can be greatly affected by the initial hydrogen occupancy at the GBs.

Moreover, other mechanisms, particularly hydrogen-enhanced local plasticity, can also facilitate IGC fracture due to dislocation interaction with GBs, which alters the local stress and strain states, GB structure, and hydrogen distributions through dislocation accommodation, pile up, and penetration through GBs[34,35]. In some cases, the involvement of dislocations can even become a decisive factor for IGCs and act as the core HE mechanism. Thus, we directly measured the 3D strain distributions ahead of the crack tip by tracking the movement of numerous S phase ($Al_2MgCu$) particles within the tensile material using the synchrotron-based imaging and microstructural features tracking techniques (see Methods), enabling nondestructive examination of the influence of hydrogen on the local plasticity at reasonable temporal and spatial resolutions. Figure 3a, c show the gradual accumulation of plastic strains in the LT material, which occurs in the regions both near and away from the crack tip, whereas the relative strain map (Fig. 3d) indicates obviously enhanced local straining at the crack tip when the macroscopic strain increases from 8.9 to 14.0%. With crack growth, the stress-driven hydrogen diffusion toward the crack tip can lead to enhanced dislocation mobility due to the hydrogen shielding effect[4], provided sufficient hydrogen atoms are attached to dislocations,

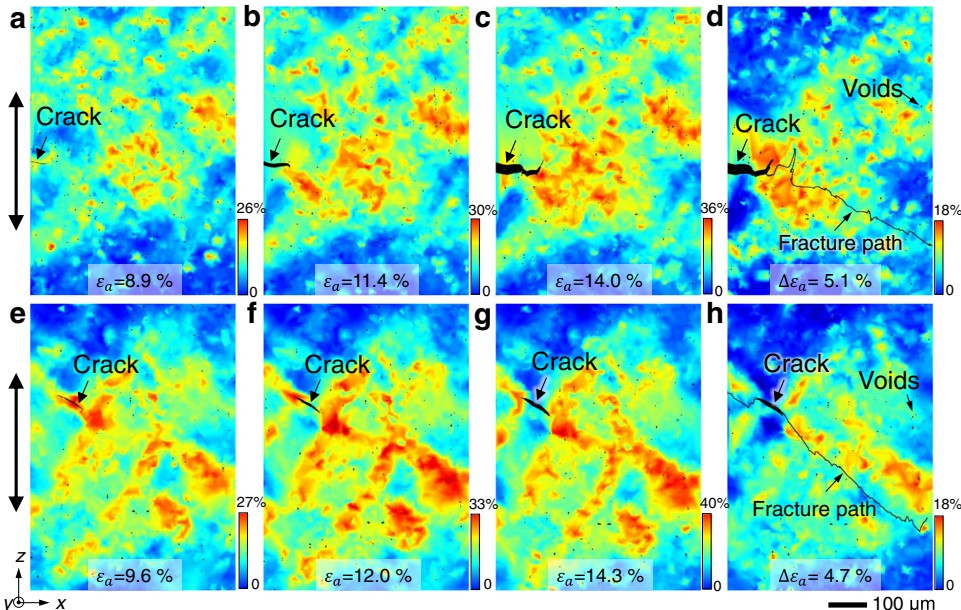

**Fig. 3 | Evolution of strain distributions around the crack tip in LT and HT materials. a–c** High-density equivalent strain ($\varepsilon_{eqv}$) maps in the virtual x-z cross sections of the LT material at an applied strain of $\varepsilon_a$ = 8.9, 11.4, and 14.0%, respectively, obtained from 3D particle tracking. **d** Relative equivalent strain map between $\varepsilon_a$ of 8.9% and 14.0%, indicating severe strain localization at the tip of the main crack. **e–g** Equivalent strain maps in the virtual x-z cross sections of the HT material at an applied strain of $\varepsilon_a$ = 9.6, 12.0, and 14.3%, respectively. **h** Relative equivalent strain map between $\varepsilon_a$ of 9.6 and 14.3%, in which suppressed strain localization at the crack tip is observed due to the change in precipitates.

forming a protective atmosphere that allows them to move more easily in certain directions. Despite the debate on HE mechanisms, in terms of either how hydrogen interacts with dislocations[36] or under what critical conditions the dislocation-based mechanism can dominate the final IGC fracture[37], a lower HE sensitivity is anticipated if the hydrogen concentration at dislocations can be reduced. This is observed in the T phase-rich material, shown in Fig. 3e, h, indicating a strain distribution throughout the whole shear bands, instead of being concentrated at the crack tip.

It is anticipated that the strong hydrogen trapping at the T phase effectively reduces hydrogen concentrations at other trap sites, including dislocations, GBs, vacancies, and nanosized η precipitates, which were shown to be sensitive to HE. The reduced hydrogen coverage at dislocations and GBs near the crack tip, due to the presence of numerous nanosized T precipitates, is expected to weaken hydrogen-enhanced local plasticity and, accordingly, its contribution to hydrogen-induced debonding at GBs. As a result, the cohesive energy of the GB can maintain at a sufficiently high level above the critical value required for separation. The whole process, which occurs in a limited nanoscale area[27] near the GB, strongly depends on the local hydrogen partitioning among various hydrogen trap sites, including dislocations, vacancies, GBs, voids, and precipitates. This nanoscopic hydrogen partitioning behavior during crack growth was predicted, based on the experimentally measured 3D distributions of various trap sites, providing real insights into the role of the T phase in hydrogen trapping. In the plastic zone in front of the crack tip, various trap sites were visualized in 3D and incorporated into the calculation of 3D hydrogen distributions, which were established based on the thermal equilibrium among these trap sites. We focused on the regions with a size of 80 × 80 × 80 µm³ ahead of the main crack at an applied strain of 14.0–14.3% (Fig. 4a, c). Severe and complicated hydrogen-trap interactions are reasonably expected to occur within the plastic zone around the crack tip, particularly near the point with peak stress and hydrogen concentration. The stress localization can assist preferential particle damage and, accordingly, void initiation and growth in front of the crack, as can be observed in the 3D distributions of particles and

micron-sized voids shown in Fig. 4a, c, which indicates predominant particle breakage in the plastic zone, whereas no obvious change in the void distribution caused by hydrogen was found. Simulations indicate hydrogen trapping in the interior of the S phase (see Methods), implying that the effect of hydrogen on IGCs by accelerating interfacial decohesion at particles is limited. Furthermore, the dislocation density distributions, including both the statistically stored dislocations, proportional to the equivalent plastic strain, and the geometrically necessary dislocations, proportional to the gradient of the equivalent plastic strain, were obtained from strain maps (Fig. 4b, d) and both of these kinds of dislocations have been suggested to affect HE by interacting with hydrogen at the crack tip[38].

The distributions of the hydrogen concentration and occupancy at dislocations, GBs, vacancies, precipitates, particles, and microvoids in the material interior were quantitatively assessed from local partitioning calculations. Figure 4e shows that in the presence of the T phase, the majority of hydrogen atoms go to the T phase and voids in both undeformed regions and the plastic zone at the crack tip, which is believed to be the main cause of crack growth suppression. Notably, all the trap sites were included in the hydrogen partitioning calculation, with the error bars originating from the heterogeneous distribution of trap sites. The only factor that is difficult to inherently incorporate into the calculation is the stress effect on hydrogen diffusion due to the complex heterogeneous distribution of strong trap sites within the stress field. Nonetheless, we reasonably expect that the T-phase nanoprecipitates remain effective in hydrogen trapping even in the nanoscale highly stressed regions. Unlike the plate-shaped η phase and coarse intermetallic particles (such as the previously reported Fe-rich[30] and Mn-rich[25] particles, also shown to be effective in hydrogen trapping), the small size and spherical shape of T precipitates endow them with low-stress localization at sharp edges and excellent dynamic hydrogen trapping capacity in a transient hydrogen diffusion scenario. In the absence of neighboring voids, the hydrogen trapping effect in the T phase leads to a 2–3 orders of magnitude reduction in the hydrogen concentrations at dislocations, GBs, and vacancies (Fig. 4f), which improves the HE resistance through any of these likely

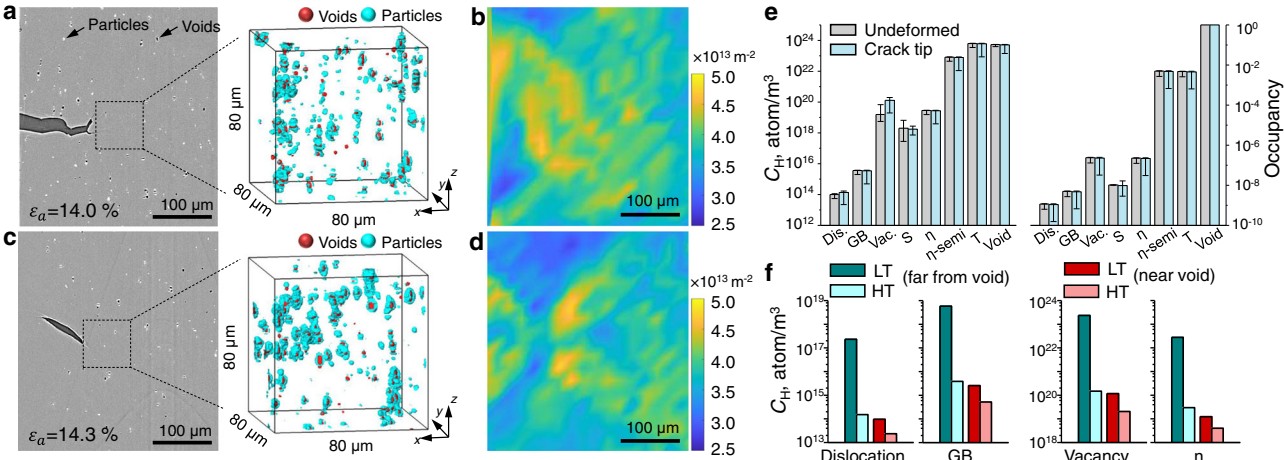

**Fig. 4 | Local hydrogen partitioning behavior around the crack tip in LT and HT materials. a** Virtual cross-section X-ray microtomography (XMT) image of the main crack at an applied strain of $\varepsilon_a = 14.0\%$ and 3D perspective view of voids and particles in the LT material. **b** Distribution of the total dislocation density at the same location as in **a**. **c** Virtual cross-section XMT image of the main crack at an applied strain of $\varepsilon_a = 14.3\%$ and 3D perspective view of voids and particles in the HT material. **d** Distribution of the total dislocation density at the same location as in **c**. **e** Comparison of hydrogen concentrations ($C_H$) and occupancies at various trap sites, i.e., dislocations, GBs, vacancies, S phase, coherent η/Al interfaces, semi-coherent η/Al interfaces, T phase, and voids, in undeformed regions and ahead of the crack tip for the HT material. The bar height represents the mean value, whereas the error bars, which originate from the heterogeneous distribution of trap sites, mark the upper and lower bounds. **f** T phase-induced reduction in the hydrogen concentrations at dislocations, GBs, vacancies, and coherent η/Al interfaces in the plastic zone ahead of the crack tip, with the blue bars showing the regions far from voids and the red bars showing those adjacent to voids.

pathways. In terms of dislocations, although hydrogen-enhanced dislocation nucleation and movement have been repeatedly captured by nanoscale in situ observations, typically in alloys lacking deep hydrogen traps[39,40], it is believed that the hydrogen-enhanced local plasticity can be largely suppressed in the present case due to the richness of strong nanoscopic hydrogen traps, which instantly drive hydrogen diffusion away from dislocations despite the fast stress variation at the crack tip. In terms of vacancies, a large quantity of vacancies is generally considered to always be generated with the creation of dislocations, and vacancies occupied with a sufficiently high amount of hydrogen can favorably aggregate to form platelets on a specific crystallographic plane, acting as embryos for microvoids and cracks[5]. In terms of η phase precipitates, spontaneous debonding at the coherent η/Al interfaces (with a hydrogen binding energy of 0.3 eV) may also occur due to the stress-induced high hydrogen occupancy at the edge surfaces[11], providing another likely pathway for HE, whereas no such phenomenon has been found at semi-coherent η/Al interfaces despite their high hydrogen trapping the energy of 0.56 eV[30]. All the processes occurring at dislocations, GBs, and coherent precipitate interfaces are expected to be suppressed by the strong hydrogen trapping effect of the T phase.

Our experiment-simulation combined study presents a realistic prediction of spatially and time-resolved hydrogen distributions during crack growth, providing a critical basis for clarifying the roles of different trap sites, particularly precipitates, in hydrogen-induced intergranular fracture. It is believed that the competition effect exists between the two types of strengthening nanoprecipitates: hydrogen trapping at coherent η/Al interfaces induces interfacial debonding (negative effect), while the hydrogen trapping within the T phase strongly suppresses it with a much higher binding energy (positive effect). The beneficial role of the T phase in HE suppression originates from its high hydrogen trapping capacity (high hydrogen binding energy in its interior instead of interfaces, thereby high trap site density) and low-stress localization (spherical shape and small size). A noteworthy step forward from the previous works utilizing similar imaging and simulation methods is the new HE mitigation strategy proposed and verified here, through the modification of nanoscopic precipitates and transforming them into strong hydrogen traps. This

strategy is expected to be effective in various high-strength aluminum alloys due to the wide availability of the T phase and can also inspire the development of HE-resistant alloys with similar switchable nanostructures. The present study also serves as an important supplement to the roles of the T phase in enhancing the mechanical properties of aluminum alloys, such as their great potential in increasing formability[41,42] and irradiation resistance[43] reported previously.

## Methods

### Materials

The aluminum alloy used in the present study has chemical compositions in mass % of 5.6Zn, 2.5Mg, 1.6Cu, and the balance Al. The specimen preparation procedures started with casting, homogenization (460 °C for 6 h and further increased to 465 °C for 24 h), hot rolling (400 °C, 87.5% thickness reduction), and thermal cycling (TC, 500 °C for 30 min, cooled in air, repeated eight times). Then the flat specimens with cross-sectional areas of approximately 0.6 mm × 0.6 mm were cut from the sheet plate by electrical discharge machining in water. After cutting, the specimens were electropolished in acid (5% HClO$_4$ + 95% methanol, at a voltage of 12 V) for 30 s and subsequently put into a salt bath for solution treatment (ST, 470 °C for 1 h, quenched in ice water). The specimens were then divided into two groups, which were aged in oil at high temperature (150 °C) and low temperature (120 °C), respectively. After that, all the specimens were aged in humid air at 120 °C for 1 h in the same container, for the purpose of hydrogen charging. After hydrogen charging, the specimens were removed from the container, cleaned, dried, and kept in acetone until tensile tests or desorption tests. The specimen preparation procedures are illustrated in Supplementary Fig. 1.

### Synchrotron X-ray microtomography (XMT) imaging

The projection-type XMT images were acquired at the BL20XU undulator beamline in SPring-8. A liquid nitrogen-cooled Si (111) double-crystal monochromator was utilized to produce a monochromatic X-ray beam with a beam energy of 20 keV. The image detector consisted of a digital CMOS camera (ORCA Flash 4.0: 2048 × 2048 pixel, Hamamatsu Photonics K.K.), a single-crystal scintillator (Ce: Lu$_2$SiO$_5$), and a lens (10×). A total of 1800 radiographs, scanning 180° with a 0.1-degree

increment, were captured in each scan. The effective pixel size of the detector was 0.5 μm, and the sample-to-detector distance was 20 mm.

### In situ tensile tests

Tensile tests were performed with a displacement rate of 0.02 mm/s using a miniature test rig (Deben UK Ltd). At each step, the displacement was held constant for 30 min before the next XMT scan. During such holding time, hydrogen redistribution occurs without significant hydrogen loss[44]. The displacement increment at each step was approximately 0.02 mm (corresponding to an applied strain of 2–3%) and the XMT images were obtained at all steps.

### Microstructural characterization

Microstructures were observed using SEM (JSM-IT800) equipped with a dispersive X-ray spectrometry (EDS) detector and spherical aberration-corrected scanning transmission electron microscope (STEM, JEOL ARM-200F) combined with a HAADF detector. For grain size analysis, the flat specimens were etched with a 2.5% $HNO_3$ + 1.5% HCl + 1.0% HF + 95% $H_2O$ solution for 15 s, washed with acetone, and observed under an optical microscope (DSX 500). APT analyses were performed on a CAMECA Instrument Inc. Local Electrode Atom Probe (LEAP) 5000 XR (reflectron fitted) in laser-pulsing mode (laser pulse energy of 40 pJ at a pulsing rate of 250 kHz), with the specimen at a base temperature of 50, and with five ions detected per 1000 pulses on average. APT reconstruction and analysis were performed using the CAMECA software AP Suite 6.1 and reconstructions were calibrated using the crystallographic features.

### Thermal desorption tests

Thermal desorption analysis (TDA) was performed to measure the hydrogen concentration in the specimens after hydrogen charging. The tensile specimens were used for TDA analysis, during which the temperature of the specimen was raised from room temperature to a maximum temperature of 550 °C, at a heating rate of 90 °C/h.

### Image analysis

The filtered back-projection algorithm was used to reconstruct the image slices from 1800 radiographs, from which the 16-bit images with a size of 2048 × 2048 $px^2$ and a total number of 2048 were obtained. The 16-bit images were transformed into eight-bit images using an absorption coefficient range of −30–40 $cm^{-1}$, which fits the eight-bit grayscale from 0 to 255. The coordinates, size, and shapes of $Al_2MgCu$ particles and voids were quantitatively analyzed using home-developed MATLAB-based programs, which segmented the images using the grayscale ranges of 200–255 and 0–100, respectively. Only features over 26 voxels in volume were analyzed to minimize the effect of noise. Following the quantitative analysis, the particle tracking was performed based on the microstructural feature tracking technique with the details described elsewhere[45,46]. In short, the same particles at different loading steps were precisely matched by comparing their gravity centers, volumes, and surface areas. This enables the generation of high-density 3D strain maps, by dividing the specimen interior into numerous tetrahedrons with the tracked $Al_2MgCu$ particles (with a total number of 55, 890) as vertices.

### Hydrogen partitioning analysis

Hydrogen atoms between lattice sites and trap sites are in a thermal equilibrium state[18]:

$$\frac{\theta_t}{1 - \theta_t} = \theta_L \exp\left(\frac{E_b}{RT}\right) \tag{1}$$

where $E_b$ is the trap binding energy, R is the universal gas constant, T is the temperature, $\theta_t$ is the occupancy at trap sites, and $\theta_L$ is the occupancy at lattice sites.

The hydrogen concentration and occupancy at all trap sites can be determined, given the total hydrogen concentration, trap densities, and binding energies of each trap site:

$$C_H^T = \theta_L N_L + \sum \theta_{ti} N_{ti} + C_{pore} \tag{2}$$

where $C_H^T$ is the total hydrogen content, $N_L$ is the trap site density at normal lattice sites, $\theta_{ti}$ is the occupancy for the $i$th trap site, $N_{ti}$ is the trap site density for the $i$th trap site, $C_{pore}$ is the hydrogen concentration at pores.

The binding energies for different trap sites can be determined by first-principles simulation: grain boundary, 0.2 eV[47]; screw dislocation, 0.11 eV[48]; edge dislocation, 0.18 eV kJ/mol[48]; vacancy, 0.3 eV[49]; pore, 0.7 eV[50]; the interior of $Al_2MgCu$ particles, 0.22 eV[51]; coherent $MgZn_2$ interfaces, 0.08–0.35 eV[52]; semi-coherent $MgZn_2$ interfaces, 0.56 eV; the interior of T phase, 0.56 eV (second-highest binding energy).

The trap densities of different trap sites ($N_{ti}$) are determined as follows: $N_{t\,GB} = 7.3 \times 10^{23}$ sites/$m^3$ for GB, based on the grain size of 90 μm in the present equiaxed materials; the number density, size and shape of η and T precipitates were measured from high-resolution TEM images (Fig. 1 and Supplementary Fig. 2), giving an average diameter of 3.9 nm and thickness of 1.8 nm in LT specimen, an average diameter of 6.2 nm for both η and T and thickness of 2.1 nm for η in HT specimen, based on which the trap densities were determined as $N_{t\,η} = 7.7 \times 10^{25}$ sites/$m^3$ in LT specimen, $N_{t\,η\,coherent} = 1.25 \times 10^{26}$ sites/$m^3$, $N_{t\,η\,semi} = 1.63 \times 10^{25}$ sites/$m^3$ and $N_{t\,T} = 1.32 \times 10^{26}$ sites/$m^3$ in HT specimen; the values of $N_{ti}$ for dislocations and vacancies were determined based on the equivalent strains in the 3D strain maps; for $Al_2MgCu$ particles and pores, the $N_{ti}$ was calculated based on the 3D statistics (including coordinates, diameter, volume of each particle or pore) obtained from the geometric quantitative analysis of XMT images.

The analyzed regions were divided into numerous cubic cells with a size of 20 μm and it is assumed that the thermal equilibrium state is reached within these cells. On this basis, hydrogen partitioning calculations were performed for different regions of interest, including the undeformed regions (far from the crack tip) and the regions near the crack tip. At such a length scale, the nanosized precipitates can be assumed to be uniformly distributed, whereas the heterogenous distributions of other trap sites such as dislocations, micro-pores, and S phase particles lead to the variance in the calculated results in different unit cells.

### First-principle calculation

First-principles calculations were conducted within the density functional theory framework using the Vienna ab initio simulation package[53] with the Perdew−Burke−Ernzerhof generalized gradient approximation exchange-correlation density functional. The Brillouin-zone k-point samplings were chosen using the Monkhorst−Pack algorithm, where a 3 × 3 × 3 k-point was used in the calculation model. A cut-off in plane-wave energy of 360 eV was applied using a first-order Methfessel−Paxton scheme with a smearing parameter of 0.2 eV. The total energy was converged within $10^{-6}$ eV/atom for all calculations. The relaxed configurations were obtained using the conjugate gradient method that terminated the search when the force on all atoms was reduced to 0.01 eV/Å. The zero-point energy (ZPE) of hydrogen atoms was not considered for the total energy because no significant effect of ZPE correction was observed in the preliminary calculations, as shown in Supplementary Table 1. Atomic configurations were visualized using VESTA 3.4.4.

The crystallographic structure of the T phase with lattice parameters of a = b = c = 1.416 nm (bcc structure, Im$\bar{3}$m space group) was used in the simulation. Two kinds of deep hydrogen trap sites were found within T precipitates: the first one has a maximum energy of 0.6 eV with a relatively low trap site density of 12 sites/unit cell; the

other one exhibits a second-highest trap energy of 0.56 eV, with considerably higher trap density of 24 sites/unit cell than the previous one.

## Data availability

The experimental and computational data that support the findings of this study are available from the corresponding authors on request. The thermal desorption, tensile test, and SEM data used in this study are available in the database under accession code https://1drv.ms/u/s! AujHvlECIhSVxQphZnkljvBmGeix?e=AVksxq.

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

## Acknowledgements

We thank the Japan Synchrotron Radiation Research Institute for supporting the synchrotron radiation experiments at SPring-8 through proposal number 2020A1796/1084. We thank Ms. Chiharu Koga for the assistance in image acquisition at Spring-8, and Mr. Yuki Fukuda for performing the thermal desorption analysis. We thank Ms. Huihui Zhu at the University of Science and Technology Beijing for her contribution to the APT experiments. H Toda acknowledges the financial support from the Japan Science and Technology Agency through Core Research for Evolutional Science and Technology (CREST) project (grant JPMJCR-1995) and the Japan Society for the Promotion of Science (JSPS) through the KAKENH project (grant JP21H04624).

## Author contributions

H.T. conceived the idea and designed the experiments. Y.W., B.S., and H.F. prepared the specimens and performed synchrotron-based in situ tensile tests. Y.X. performed the TEM observations. Y.W., B.S., and H.F. analyzed the CT images and obtained 3D strain distributions. Y.W. and K.S. performed hydrogen partitioning calculations. K.H., A.T., and M.U. assisted in the experiments at the synchrotron radiation facility in Spring-8. Y.W. prepared the original manuscript with input from all authors. H.T. revised the manuscript. All authors have given approval for the final version of the manuscript.

## Competing interests

The authors declare no competing interests.
