## [Peer Review File · Nature Communications]

Switching nanoprecipitates to resist hydrogen embrittlement in high-strength aluminum alloysREVIEWER COMMENTS

Reviewer #1 (Remarks to the Author):

The issue of HE in high strength materials has been a topic of interests for centuries now; and this work is no exception.

I commend the authors for this comprehensive work on the HE of Al. The research is original, and your combination of simulation and experimental work is deserving praise as it is well-thought out. There was a conscientious effort to present results effectively and provide convincing evidences to arguments. Data analysis and interpretation is logical; and the conciseness of the work is also meritorious.

Your work presents a significant finding related to HE mitigation and will be useful in terms of providing insights to future mitigation techniques for other metals.

Please see attached for some minor comments and suggestions to help improve the quality of the work.

Reviewer #2 (Remarks to the Author):

The manuscript report the T phase based Al alloy to enhance HE resistance. I have a few questions below

1. The authors claim the ability of T phase to H trapping from DFT simulations. It will be more convincing to compare such kind of results with other precipitates for example MgZn₂. Moreover, the authors claim no need to include zpe. More evidence is needed. More over, the entropy will also be involved in addition to zpe, and how the binding energy change with the H concentration is also of interests to see the real capacity to trap H.
2. How about T phase compared to other precipitates other than the η' phase?
3. It is also not clear how the interface between Al and precipitate affect the H trapping and diffusion. For example, coherent, incoherent interface and strain's effects.
4. What is the general guidance to the design HE resistance alloy? It will be better the authors could elaborate on that.

Reviewer #3 (Remarks to the Author):

Revision

Comments to the authors:

This paper is extremely interesting.

In fact, I did not see any mention in the manuscript, but the AlMgZn(Cu) alloys, which is the metallurgical merge between 5xxx and 7xxx series of Al-based alloys, comprise an entire new class of alloys, which has been discovered just in the past 3 years: the aluminium crossover alloys. A recent review paper has been published on the topic and I suggest the authors to read it and possibly incorporate these ideas into their manuscript, if they find it appropriate and suitable (<https://doi.org/10.1016/j.pmatsci.2021.100873>).

Interestingly, one of the major features of novel aluminium crossover alloys is the precipitation of the T-phase – Mg₃₂(Zn,Al)₄₉ – which is a highly chemically-complex intermetallic phase and highly-concentrated superstructure bearing hundreds of atoms in its unit cell; this is opposed to most conventional hardening precipitates (and not nanoparticles as the terminology used by the authors in their paper) in existing commercial aluminium alloys.

That said, I will now make some comments on the authors' paper: this is a very good piece of scientific research, but the excessive number of supplemental figures makes it almost impossible to follow its intended message by only reading the main manuscript text. I suggest the authors to re-evaluate the use of supplemental figures by incorporating the most relevant figures (i.e. those figures who are fundamental for major understandings) into the main paper.

Comments on figure 1: this reviewer is not sure about the indexing of both eta-prime and T-phase using the SAED and FFT method along [110] zone axis. It is of my knowledge that these phases can be unequivocally distinguished along the [001] zone axis instead, please see these two references:

<https://doi.org/10.1016/j.actamat.2020.116617>

<https://doi.org/10.1016/j.matdes.2020.108837>

In addition, for an article of the Nature Communication's relevance, this reviewer finds it mandatory that the authors present some sort of chemical analysis of these precipitates: either APT or STEM-EDX. These phases can be very similar on a chemical nature standpoint, and unequivocal confirmations of their existence are mandatory for validate the main message of the paper, which is the hypothesis that T-phase precipitates contribute to a significant reduction in the concentration of hydrogen within an Al-based alloy (a new crossover alloy type synthesized by the authors?). In some aluminium crossover alloys, the diffraction signal of T+eta-prime phases come together, but there are ways to distinguish them; I advise the authors to consult appropriate literature.

Now a comment on semantics. Can a nanoparticle be considered a precipitate and vice-versa? Well, precipitates (i.e. a secondary phase that separate/partition out the alloy's matrix whilst still embedded to it) via natural and artificial aging are known to occur in aluminium alloys since 1938 (Nature 142, 569-570 and Nature 142, 570-570 both from the year of 1938!). The term "nanoparticles" gained only recent attention due to the emergence of nanotechnology as a consolidated field of research. Personal opinions in paper revisions can cause stress to the authors, but it is the subjective opinion of this reviewer that a precipitate is not "technically" a nanoparticle. The authors have not dispersed nanoparticles into an Al-based alloy, rather produced these precipitates via conventional and well-known metallurgical methods. I tend to view nanoparticles as nanomaterials that are not necessarily are embedded into a matrix. Some philosophical comments for the authors to think about!

Precipitate coherency is mentioned in line 95, but no coherence study between the alloy and the precipitates has been performed. Similarly, the authors mention in lines 108-109 those mechanical properties are significantly improved "(...) due to the change in nanoprecipitates (...)", but no clarification has been provided.

Micrographs in Fig. 2F and 2J are too small and of low resolution to detect any intergranular crack. Fig. 2 needs to pass a complete review and redesign.

When the authors say "(...) at the same hydrogen content level (...)", this reviewer thinks: how precise are the existing experimental methods for hydrogen detection in metals? Are they so precise one can claim levels are the same?

Fig. 3 is clear and good. Increase font size will help Nature Communication's readership to better grasp the figure's message.

On Fig. 4, plots 4E and 4F, I miss some error bars and a critical evaluation of the systematic and statistical errors that may arise during the measurement of hydrogen concentrations. This is of paramount importance to substantiate the author's nice discovery!

I would like to give another opportunity for the authors to streamline all the discussions presented between lines 135-236. By streamlining this review means that the authors should: (i) stick with their discovery; (ii) answer clearer why T-phase precipitates are better than eta-prime (and others) to remove hydrogen interstitials from the alloy's matrix; and (iii) the implications of their discovery faced by previous literature.

One point is missing in the authors discussion though. Recently, it has been reported that T-phase precipitates are able to resist impact to energetic particle irradiation in a novel aluminium crossover alloy (<https://doi.org/10.1002/adv.202002397>). It has been reported that the T-phase is able to survive higher doses than other hardening phases in commercial aluminium alloys (i.e. Mg₂Si in 6xxx series Al-based alloys). One of the explanations provided by the authors to explain such a radiation tolerance is the fact that in aluminium crossover alloys, the volumetric fraction of T-phase precipitates is significantly higher than other phases in commercial aluminium alloys. If the number of precipitates divided by the volume is higher, then they act as a powerful sink for irradiation-induced defects. This reviewer question if the same thing happens here in this discovery reported by Y. Wang et al. in this present submission? Perhaps, the enhanced resistance to hydrogen embrittlement arises since upon precipitation of T-phase precipitates, the precipitation density increases considerably compared to eta-prime, leading to higher sinking efficiency for hydrogen.

Dear reviewers

We appreciate all the efforts spent in reviewing our manuscript and are really thankful for the constructive comments. As required by the reviewers, we performed additional experiments and analyses to further improve the quality of this manuscript.

The point-by-point response is listed as follows. All the changes were marked in **yellow background** in the revised manuscript.

Reviewer #1 (Remarks to the Author):

The issue of HE in high strength materials has been a topic of interests for centuries now; and this work is no exception.

I commend the authors for this comprehensive work on the HE of Al. The research is original, and your combination of simulation and experimental work is deserving praise as it is well-thought out. There was a conscientious effort to present results effectively and provide convincing evidences to arguments. Data analysis and interpretation is logical; and the conciseness of the work is also meritorious.

Your work presents a significant finding related to HE mitigation and will be useful in terms of providing insights to future mitigation techniques for other metals.

Please see attached for some minor comments and suggestions to help improve the quality of the work.

Response: We really appreciate the recognition from the reviewer and are glad to hear that we are on the right track.

Comment: I am a bit concerned with the corrugated shape of the stress-strain curve in the plastic region (Fig. 1A). Can you explain what caused this?

Response: As explained in the Materials and Methods in supplementary materials, quasi in-situ tensile tests were performed in the present study to obtain a series of 3D images of the Al alloys. In this sort of interrupted tensile tests, the specimen was hold still for a period of time (30 mins in present study), during which thousands of CT images were acquired. The corrugated shape of stress-strain curve is due to the **stress relaxation** that naturally occurs during specimen holding. The interrupted tensile test has been widely utilized to study 4D evolution behavior of microstructures within materials, by combining it with X-ray

tomography, see some examples:

<https://doi.org/10.1016/j.actamat.2022.117658> (Fig. 1)

<https://doi.org/10.1016/j.actamat.2017.01.010> (Fig. 2)

<https://doi.org/10.1016/j.polymertesting.2022.107551> (Fig. 1)

These examples include the previous studies on hydrogen embrittlement of aluminum alloys in the author's laboratory.

However, there are indeed some disadvantages of this interrupted tests although they are popular and useful in the 4D observations, possibly including those issues that concern the reviewer. One is that it becomes tricky to get the accurate fracture strain from the serrated stress-strain curves due to the small size of specimen (no strain gauge was used for such small specimen). To solve this problem, the true strains in each step were corrected by precisely measuring the displacements of different pairs of particles within the specimen during plastic deformation. The strain values between different two neighboring steps were calculated by linear interpolation. The fracture strain was predicted by linear extrapolation based on the two strain values in preceding two steps. Therefore, the corrugated shape does not affect the measurement of the elongation loss, which has been used as an indicator of hydrogen embrittlement.

Comment: Doubt on the usage of 'Fractional' in Fig. 2b.

Response: We intended to say the areal fraction of crack on the fracture surface by 'Fractional area'. To avoid confusion, 'Fractional area' was replaced with 'Areal fraction'.

Comment: (Fig. 3) maybe mention when these maps where derived? meaning what technique

Response: We agree that the techniques used for obtaining the 3D strain maps in Fig. 3 should be described more clearly in the text. The high-density strain maps were generated by tracking the movement of numerous particles at different loading steps within the specimen, using the MATLAB-based image segmentation and tracking algorithms developed in the authors' lab, i.e., the so-called microstructural feature tracking (MFT) technique.

In short, the same particles (in this case the micron-scale Al_2MgCu particles) at different loading steps were precisely matched by comparing their gravity centers, volumes and surface areas. This enables the generation of high-density 3D strain maps, by dividing the specimen interior into numerous tetrahedrons with the tracked Al_2MgCu particles (with a total number of 55, 890) as vertices.

The following steps are necessary in the mapping of 3D strains:

1. **Quantitative geometric analysis.** Polygonization techniques such as the marching cubes algorithm introduced serve as the basis of 3D image analysis. Volume, surface area, and length measurements can be accurately conducted once the surface morphology has been determined. The MATLAB-based home-made software used by the authors is capable of analyzing various microstructure parameters, including those used for measuring the size, shape, and spatial distribution.
2. **Particle tracking.** First, an affine transformation was applied to register the 3D images in two loading steps, which is the key to the success of following tracking analysis. The matching probability parameter (M_p) was defined to evaluate the matching accuracy of particles before and after deformation, by comparing the gravity center, volume and surface area of these two particles, and a threshold M_p value was applied to filter out the mis-matched particles.
3. **Strain mapping.** The sample interior can be divided into multiple tetrahedrons with the tracked particles as vertices by using the Delaunay tessellation. Assuming the deformation is small, the vertical strain ($\epsilon_x, \epsilon_y, \epsilon_z$) and shear strain ($\gamma_{xy}, \gamma_{yz}, \gamma_{zx}$) in a single tetrahedron can be calculated from the particle deformation (u_i, v_i, w_i), (u_j, v_j, w_j), (u_k, v_k, w_k), and (u_l, v_l, w_l) at four vertices i, j, k , and l .

This process can be illustrated in **Fig. R1** shown below ('X-ray CT', Hiroyuki Toda, Springer, 2022, Chapter 9. 4D image analysis, Fig. 9.3):

Fig. R1. Schematic explaining 4D image analysis conducted by finding an identical characteristic point in continuous 3D images, conducting particle tracking for all particles, and using these results to conduct 3D mapping of various mechanical quantities

More details of this technique can be found elsewhere:

[1] 'In-situ 3D observation of hydrogen-assisted particle damage behavior in 7075 Al alloy by synchrotron X-ray tomography', Wang, Toda et al., Acta Mater., 2022.

<https://doi.org/10.1016/j.actamat.2022.117658>

[2] 'High-density three-dimensional mapping of internal strain by tracking microstructural features', Kobayashi, Toda et al., Acta Mater., 2008.

<https://doi.org/10.1016/j.actamat.2007.12.058>

[3] Toda, *X-ray CT- Hardware and Software Techniques*, Springer Nature, 2021. (Chapter 9-'4D image analysis', Page 496-517). <https://link.springer.com/book/10.1007/978-981-16-0590-1>

Page 6 in revised manuscript: '*Thus, we directly measured the 3D strain distributions ahead of the crack tip by tracking the movement of numerous Al₂MgCu particles within the tensile material using the synchrotron-based imaging and microstructural features tracking techniques*'.

Comment: lack of description for Fig. 3(H) in figure caption.

Response: This was a pagination issue. Description for Fig. 3(H) was shown in the next page in the previous manuscript.

Comment: (comment originally marked in PDF, copied here). 'we directly measured the 3D strain distributions ahead 141 of the crack tip based on the movement of numerous Al₂MgCu particles (see fig. S6) within the 142 tensile material' . I don't see 'the movement of numerous Al₂MgCu particles' in fig. S6.

Response: Sorry for the confusion. Fig. S7 (mistakenly written as fig. S6) in the first-edition manuscript was intended only to identify the type of particles by EDS. The movement of particles was not visualized in the present case. In the manuscript, the figures were reorganized and the figure numbers were carefully checked.

In the calculation of 3D strain maps, the quantitative data (so-called object data) including the X/Y/Z coordinates of gravity center, equivalent diameter and sphericity of Al₂MgCu particles in text files were imported into the program. Then the strain maps were automatically generated without the necessity of visualizing the movement of particles, although tracking the particle movement is the principle of the strain mapping technique.

Reviewer #2 (Remarks to the Author):

The manuscript report the T phase based Al alloy to enhance HE resistance. I have a few questions below

1. The authors claim the ability of T phase to H trapping from DFT simulations. It will be more convincing to compare such kind of results with other precipitates for example $MgZn_2$. Moreover, the authors claim no need to include zpe. More evidence is needed. More over, the entropy will also be involved in addition to zpe, and how the binding energy change with the H concentration is also of interests to see the real capacity to trap H.

Response: We agree that the comparison of hydrogen trapping at T phase with other precipitates is important to understand the main discovery in the present study.

In our previous study, the H trapping at $MgZn_2$ was thoroughly investigated and discussed [*First-principles study of hydrogen segregation at the $MgZn_2$ precipitate in Al-Mg-Zn alloys*, Tsuru et al., *Compt. Mater. Sci*, 2018, 148: 310-306; <https://doi.org/10.1016/j.commatsci.2018.03.009>]. It is shown in the following **Fig. R2**:

Fig. R2. Hydrogen trap energy inside $MgZn_2$ precipitates, indicating fairly low trap energy.

It was found that H is mainly trapped at the coherent $MgZn_2/Al$ interfaces rather than the interior of precipitates, because the **hydrogen trap energy at coherent interfaces**

(0.3 eV) is much higher than that inside the precipitates (0.05 eV). It was also demonstrated that hydrogen trap energy at **semi-coherent MgZn₂ interfaces was 0.56 eV** (Wang, Toda, et al., *Acta Mater.*, 2022, 227, 117658, <https://doi.org/10.1016/j.actamat.2022.117658>).

Variants of η phase. There are too many (more than 10) variants of η phase, e.g., η' , η_1 , η_2 , η_3 , η_4 , ..., which are both structurally and compositionally complex (Chung, et al., *Acta Mater.*, 2019, 174, 351-368; <https://doi.org/10.1016/j.actamat.2019.05.041>). It remains challenging to consider the hydrogen trapping at all these different variants. In the present study, **they are unified as ' η phase'** and the simulation results for η -MgZn₂ precipitates have been utilized in this study. Same for T phase: all the variants are unified as 'T'.

In addition, we also simulated the hydrogen trap energies at many other intermetallic compound particles such as Al₇Cu₂Fe, Mg₂Si etc., although they are not found in the present study. In these studies, it has been shown that the **ZPE correction energy was within ± 0.05 eV/atom, which is negligible.** (*Hydrogen Trapping in Mg₂Si and Al₇FeCu₂ Intermetallic Compounds in Aluminum Alloy: First-Principles Calculations*, Yamaguchi et al., *Mater. Trans.*, 2020, <https://doi.org/10.2320/matertrans.MT-M2020201>). The same conclusion was reached for T phase in the present study, as shown in the **Table R1** below:

Table R1 Effect of ZPE on the calculated hydrogen trap energies at T phase.

Site	Trap energy (eV/atom)	
	With ZPE	Without ZPE
A	0.603	0.579
B	0.556	0.539
C	0.338	0.31
D	0.305	0.293
E	0.239	0.215
F	0.192	0.175
G	0.133	0.129
H	0.005	-0.041
I	-0.055	-0.075

J	-0.104	-0.13
K	-0.165	-0.195
L	-0.194	-0.226

Thirdly, the change of trap energy with multiple hydrogen atoms has also been studied, as shown in **Fig. R3** below, which indicates that T phase with 24 crystallographically equivalent trap sites in its unit cell is capable of trapping a considerable amount of hydrogen atoms.

Fig. R3. The variation of trap energy when multiple hydrogen atoms are gradually added into the model in the first principles simulation.

In the revised manuscript, these points were emphasized. See main text in Page 8 and Supplementary Table 1:

‘Moreover, spontaneous debonding at the coherent interfaces of the η phase (with a maximum hydrogen binding energy of 0.3 eV) may also occur...’

2. How about T phase compared to other precipitates other than the η' phase?

Response:

Hydrogen trapping energies at numerous types of particles were analyzed in the authors' lab, as shown in the figure **Fig. R4** below (Xu et al., Acta Mater., 2022; <https://doi.org/10.1016/j.actamat.2022.118110>) .:

Figure R4 Summary of trapping energies at various intermetallic particles

These particles include the common precipitates in Al alloys, such as Mg₂Si, Al₇Cu₂Fe, Al₂MgCu (S phase in the present manuscript) and so on. For some of these precipitates, hydrogen is trapped at interfaces, such as Mg₂Si and MgZn₂; for others, hydrogen can be trapped at the interior, such as Al₂MgCu, Al₇Cu₂Fe, and Al₁₁Mn₃Zn₂. Hydrogen trapping energy of T phase is higher than most of these particles. The Al₇Cu₂Fe, and Al₁₁Mn₃Zn₂ particles with very high binding energy are believed to be similar with or more effective in hydrogen trapping than T phase, but due to their large size (which causes significant stress localization and thereby likely defect initiation) they are considered less effective in suppressing hydrogen assisted crack growth than T phase. In conclusion, **we consider T phase as the best candidate** among these precipitates for hydrogen embrittlement suppression.

These issues including have been briefly discussed in the previous manuscript. See introduction part:

'One step forward in this direction is the newly reported beneficial effects of intermetallic compound particles in trapping hydrogen and reducing quasi-cleavage cracks in high-strength aluminum alloys ^{24, 25}. Most coarse particles are intrinsically brittle (25), and their dynamic hydrogen trapping ability is weakened by their large size since full trapping of hydrogen atoms within μm -sized particles in a short period of time is difficult...'

3. It is also not clear how the interface between Al and precipitate affect the H trapping and diffusion. For example, coherent, incoherent interface and strain's effects.

Response: For **T phase**, H is trapped **within** the precipitates, according to the FPS simulations. It is thereby believed that the effect of interface coherency on the hydrogen trapping around T phase is limited. For **η phase**, previous studies indicated that small-sized precipitates exhibit coherent interfaces, whereas large precipitates have semi-coherent interfaces. To confirm this, **additional HRTEM observations were performed**, which indicated mismatch dislocations at the relatively coarse MgZn_2 precipitates in high-temperature (HT) aged material in this study, as shown in **Supplementary Fig. 4** below.

Supplementary Fig. 4. HAADF-STEM image of the interfaces at MgZn_2 precipitates in HT material, indicating semi-coherent interfaces at the side surfaces.

Coherent η/Al interfaces trap H with a relative lower binding energy (**0.3 eV**), but H atoms strongly decreases the binding energy until zero, i.e., the hydrogen induced

spontaneous debonding at coherent η/Al interface reported by Tsuru et al. (<https://www.nature.com/articles/s41598-020-58834-6>). Semi-coherent η/Al interfaces trap H with higher binding energy (0.56 eV), but no such spontaneous debonding has been found. The 3D interior of T phase traps hydrogen with a high binding energy (with a maximum of 0.6 eV), and they trap more H than 2D semi-coherent η/Al interfaces. Even in the presence of some fractions of semi-coherent η/Al interfaces in HT material, T phase is still effective in H trapping. Specifically, as shown in the **revised Fig. 4e**, hydrogen concentration at T phase is one order of magnitude higher than that at semi-coherent η/Al interfaces.

In the first-edition manuscript, all the interfaces at MgZn_2 were assumed to be coherent. In the revised one, **all the side interfaces at the MgZn_2 in HT material were assumed to be semi-coherent**. This is a conservative assumption for estimating the trapping effect of T phase, keeping in mind that semi-coherent interfaces at MgZn_2 trap more hydrogen than coherent ones because of their high binding energy (0.56 eV), and the detrimental effect of hydrogen-assisted interfacial debonding at MgZn_2 . In other words, using this assumption, we estimate the lowest amount of hydrogen trapped by T phase, in the competitive hydrogen trapping scenario between MgZn_2 and T phase.

In terms of strain's effect, it is believed that the spherical-shaped T phase exhibits **much lower stress localization** than that at the edges of elongated MgZn_2 . Therefore, it is reasonable to assume that the T phase suppresses hydrogen embrittlement by reducing the risks of crack initiation around MgZn_2 phase, which is expected to be strongly related to the stress/strain distributions at its edges.

More discussions about these issues have been added into the revised manuscript.

4. What is the general guidance to the design HE resistance alloy? It will be better the authors could elaborate on that.

Comment: It is believed that the design of strong and HE-resistant Al alloy should overcome the conflict between strength and HE. **The understanding and utilization of H interaction with main strengthening element** is an important basis. For 7XXX alloys, MgZn_2 precipitates act as strengthening precipitates but existing studies indicate hydrogen induces spontaneous debonding at their interfaces. To suppress HE initiated from such nano-scale H-trap interactions, a strong nano-sized hydrogen traps (precipitates) should be introduced to reduce hydrogen concentration at the defects. An ideal type of precipitates is that they strongly trap H without being embrittled by the H themselves or inducing stress localization around their edges. We reported such effects for T phase in the present study: **high H trapping capacity** (high trapping energy, and H is trapped inside instead of interfaces, thereby high trap site density); **round shape and small size** endow them with low stress localization. These issues are more clearly summarized in the conclusion part.

Reviewer #3 (Remarks to the Author):

This paper is extremely interesting.

In fact, I did not see any mention in the manuscript, but the AlMgZn(Cu) alloys, which is the metallurgical merge between 5xxx and 7xxx series of Al-based alloys, comprise an entire new class of alloys, which has been discovered just in the past 3 years: the aluminium crossover alloys. A recent review paper has been published on the topic and I suggest the authors to read it and possibly incorporate these ideas into their manuscript, if they find it appropriate and suitable (<https://doi.org/10.1016/j.pmatsci.2021.100873>).

Response: We appreciate the constructive comment. Although the alloy class in the suggested paper (crossover alloys with **low Zn/Mg ratios**) is different from ours (7XXX, with **a Zn/Mg ratio > 1**), we found some ideas are interesting in the paper and they can be incorporated into our manuscript.

This reference pointed out that these crossover alloys exhibit strong potential to achieve better trade off between strength and other properties, i.e., formability in their case. This is well aligned with the findings in our study, i.e., T phase shows good potential to overcome strength-HE conflict. From this standpoint, the crossover alloys have attracted increasing more attention. Such trend has been added into the discussions in the revised manuscript. See the last paragraph of main text.

Interestingly, one of the major features of novel aluminium crossover alloys is the precipitation of the T-phase – Mg₃₂(Zn,Al)₄₉ – which is a highly chemically-complex intermetallic phase and highly-concentrated superstructure bearing hundreds of atoms in its unit cell; this is opposed to most conventional hardening precipitates (and not nanoparticles as the terminology used by the authors in their paper) in existing commercial aluminium alloys.

That said, I will now make some comments on the authors' paper: this is a very good piece of scientific research, but the excessive number of supplemental figures makes it almost impossible to follow its intended message by only reading the main manuscript text. I suggest the authors to re-evaluate the use of supplemental figures by incorporating the most relevant figures (i.e. those figures who are fundamental for major understandings) into the main paper.

Response: We agree with this comment and some supplementary figures, including the FPS calculation of H trapping, H content data, were moved to the main text in the revised manuscript, as shown in the **revised Figure 1 and 2**:

Revised Fig. 1. The FPS results of H trapping at T phase, and APT results for confirming T phase were added into this figure.

Revised Fig. 2. Hydrogen content measurement was added to this figure.

Comments on figure 1: this reviewer is not sure about the indexing of both eta-prime and T-phase using the SAED and FFT method along [110] zone axis. It is of my knowledge that these phases can be unequivocally distinguished along the [001] zone axis instead, please see these two references:

<https://doi.org/10.1016/j.actamat.2020.116617>

<https://doi.org/10.1016/j.matdes.2020.108837>

Response: The T phase can be identified by the diffraction patterns along either [110] or [001] zone axis, as shown below:

For **[001] zone axis**, the possible diffraction patterns in the co-existence of T and η phases can be:

Fig. R5 Illustration of the diffraction patterns for T and η phases along [001] zone axis.

(Lee et al., Journal of Japan of Institute of Light Metals, 2017, <https://doi.org/10.2464/jilm.67.162>)

Similar patterns can also be found in other references:

Zou et al., J Mater. Sci. Technol., 2021. <https://doi.org/10.1016/j.jmst.2020.12.045>

(And also, those suggested by the reviewer).

However, as shown in the above **Fig. R5**, the characteristic diffraction spots for T and η phases are very close to each other. Moreover, in the present study, the diffraction spots for T phases are too weak. So, we preferred SAED data along [110] zone axis, to clearly present the characteristic spots for T phase, without strong interference from η .

a) [001] zone axis

b) [110] zone axis

Fig. R6, SAED patterns for HT material.

The characteristic spots of T phase are located at $2/5$ and $3/5$ positions between 002 and $\bar{1}1\bar{1}$ Al spots, as shown in **Fig. R6b**. See the references below:

Wang et al., Mater. Sci. Eng. A., 2021, <https://doi.org/10.1016/j.msea.2020.140515>

Yang et al., J Alloy. Compd., 2014, <https://doi.org/10.1016/j.jallcom.2014.04.185>

The FFT analysis along [110] also provides clear evidence for T phase, as shown in **Fig. R7** below:

Fig. R7, BF image, HAADF image and FFT of T phase in the HT material

Comment: In addition, for an article of the Nature Communication's relevance, this reviewer finds it mandatory that the authors present some sort of chemical analysis of these precipitates: either APT or STEM-EDX. These phases can be very similar on a chemical nature standpoint, and unequivocal confirmations of their existence are mandatory for validate the main message of the paper, which is the hypothesis that T-phase precipitates contribute to a significant reduction in the concentration of hydrogen within an Al-based alloy (a new crossover alloy type synthesized by the authors?). In some aluminium crossover alloys, the diffraction signal of T+eta-prime phases come together, but there are ways to distinguish them; I advise the authors to consult appropriate literature.

Response: We found it is difficult to accurately measure the chemical compositions of both η and T phases using STEM-EDX, due to the small size of these two precipitates. As suggested by the reviewer, an addition group of APT tests were performed to confirm T phase. The APT results are shown in the **revised Figure 1g-i**:

Figure 1g-i APT results

APT results indicate that the total atomic percentages of Zn+Mg in many precipitates are fairly high (close to 80%), as shown in **Fig. 1h**. This is an evidence of T phase (Zou et al., Mater. Character., 2020, <https://doi.org/10.1016/j.matchar.2020.110610>). In contrast, the atomic percentage of Zn+Mg in η phase is relatively lower, around 60% (**Fig. 1i**).

Comment: Now a comment on semantics. Can a nanoparticle be considered a precipitate and vice-versa? Well, precipitates (i.e. a secondary phase that separate/partition out the alloy's matrix whilst still embedded to it) via natural and artificial aging are known to occur in aluminium alloys since 1938 (Nature 142, 569-570 and Nature 142, 570-571 both from the year of 1938!). The term "nanoparticles" gained only recent attention due to the emergence of nanotechnology as a consolidated field of research. Personal opinions in paper revisions can cause stress to the authors, but it is the subjective opinion of this reviewer that a precipitate is not "technically" a nanoparticle. The authors have not dispersed nanoparticles into an Al-based alloy, rather produced these precipitates via conventional and well-known metallurgical methods. I tend to view nanoparticles as nanomaterials that are not necessarily are embedded

into a matrix. Some philosophical comments for the authors to think about!

Response: The review proposed a really good comment about the terminology. Initially the authors changed 'precipitates' to 'particles' to fit the wide readership of this journal. Now we have decided to change back to 'precipitates'. We think the precipitates that form during quenching and aging treatment can be considered as particles, whereas the intermetallic compound particles are not necessarily precipitates (such as those particles that are purposefully added into the matrix). Although in the present study, all the second phase particles including T, η and S phases can be defined as precipitates, for clarity consideration, we refer to the nano-sized T and η phases as 'precipitates', whereas the coarse S phase as 'particles'. In this way, the 'precipitate switching' strategy proposed in the present study is only related to T and η phases.

Comment: Precipitate coherency is mentioned in line 95, but no coherence study between the alloy and the precipitates has been performed. Similarly, the authors mention in lines 108-109 those mechanical properties are significantly improved "(...) due to the change in nanoprecipitates (...)", but no clarification has been provided. Micrographs in Fig. 2F and 2J are too small and of low resolution to detect any intergranular crack. Fig. 2 needs to pass a complete review and redesign.

Response: 1. HRTEM observations of interfacial coherency at MgZn_2 precipitates in HT material were performed, as shown in **Supplementary Fig. 4:**

Supplementary Fig. 4. HAADF-STEM image of the interfaces at MgZn_2 precipitates in HT material, indicating semi-coherent interfaces at the side surfaces.

Semi-coherent interfaces were found for MgZn₂ phase in HT material. Therefore, the assumptions were modified in the hydrogen partitioning analysis (fully coherent interfaces were assumed in previous manuscript) and accordingly **Fig. 4e and f** were updated. The revised partitioning analysis indicates that some fractions of hydrogen go to semi-coherent interfaces due to their high binding energy. But this does not affect the trapping effect of T phase, because of their large trap site density (3D interior in T phase compared to the 2D interfaces at η phase).

2. The T phase leads to the improvement in ductility in the stress-strain curve in Fig. 2a and reduces the areal fractions of intergranular cracks on the fracture surfaces. In the revised manuscript, the 'improved mechanical properties' was stated more clearly.

3. Two magnified SEM images of IGC were added to **Supplementary Fig. 6**.

Comment: When the authors say “(…) at the same hydrogen content level (…)” , this reviewer thinks: how precise are the existing experimental methods for hydrogen detection in metals? Are they so precise one can claim levels are the same?

Response: The thermal desorption analysis utilized in the present study is a very precise technique, that is able to measure hydrogen content in ppb level. The hydrogen concentrations for LT and HT specimens are 774 and 783 ppb (as shown in the TDA curve in revised **Fig. 2b**), respectively. It is thereby believed that this method is accurately enough to ensure same hydrogen levels within the specimens.

Fig. 3 is clear and good. Increase font size will help Nature Communication' s readership to better grasp the figure' s message.

Response: We thank the reviewer for this suggestion. The font size was increased. The formats of all the figures were carefully checked to improve the quality.

On Fig. 4, plots 4E and 4F, I miss some error bars and a critical evaluation of the systematic and statistical errors that may arise during the measurement of hydrogen concentrations. This is of paramount importance to substantiate the author' s nice discovery!

Response: Plots 4E and 4F were updated: semi-coherent interfaces were incorporated into the calculations and error bars were added to Fig. 4e.

The analyzed regions were divided into numerous cubic cells with a size of 20 μm and it is assumed that the thermal equilibrium state is reached within these cells. On this basis hydrogen partitioning calculations were performed for different regions of interests, inclusion the undeformed regions (far from crack tip) and the regions near crack tip. At

such length scale ($\sim 20 \mu\text{m}$), the nano-sized precipitates can be assumed to be uniformly distributed, whereas the heterogenous distributions of trap site density for other trap sites such as dislocations, micro-pores and S phase particles lead to the variance in the calculated results. For example, if there is no S phase particle in the unit cell in the partitioning analysis, then the trap site density of S phase would be zero, which results in increase in hydrogen concentrations in other traps sites. However, even so, hydrogen concentration is much more sensitive to the binding energy rather than trap site density, as shown in the following figure:

Fig. R8 dependence of hydrogen occupancy at trap sites on their hydrogen trap energies

It is seen from **Fig. R8** that when the trap energy is elevated by 0.1 eV, hydrogen occupancy can be increased by around 2 orders of magnitude. Since the hydrogen concentration and occupancy in Fig. 4e are plotted in logarithmic scales, the errors do not significantly affect the evaluation on hydrogen distributions, particularly the analysis of hydrogen trapping at T phase.

For Figure 4f, since only a few cubic cells were calculated near the crack tip, we decided not to include the error bars. For example, in the case of cubic cells far from the voids (means there is no void in the cell), only **1 or 2 cells** were found in this region at crack tip. It is reasonable to assume that the errors in Fig. 4f are in the same scales with Fig. 4e, since both of them are originated from the heterogeneous distribution of trap sites. I would like to give another opportunity for the authors to streamline all the discussions presented between lines 135-236. By streamlining this review means that the authors should: (i) stick with their discovery; (ii) answer clearer why T-phase precipitates are better than eta-prime (and others) to remove hydrogen interstitials from the alloy's matrix; and (iii) the implications of their

discovery faced by previous literature.

Response: Thanks for the suggestion. We agree that the discussions provided in previous manuscript were to some extent deviated from the main logic line. This part was revised according to this comment, while at the same time the conciseness was maintained.

1. Some discussions were added to connect different paragraphs and to make sure all these parts are close related to the main topic of this manuscript. Some unrelated discussions were deleted:

~~‘Instead, particle damage occurs naturally in the form of internal breakage in a similar way to that in ductile fracture³⁹, except that the void growth is accelerated in the plastic zone due to the large strain level and high stress triaxiality.’~~

~~‘As suggested in theoretical models, a peak hydrostatic stress appears at some point in front of a crack tip, where GB decohesion is likely to occur first, under the influence of stress-enhanced diffusion toward the stressed region due to lattice expansion’~~

~~‘In contrast, the hydrogen microvoids originating from the breakage of particles, due to their high binding energy, can resist crack growth to some extent by trapping a considerable amount of hydrogen inside them as hydrogen gas²⁴’~~

2. The role of T phase is more clearly stated, including its hydrogen trapping features (i.e., the high binding energy and large volume fraction), low stress localization (i.e., spherical shape, small size), and they are in depth compared with other precipitates.

One point is missing in the authors discussion though. Recently, it has been reported that T-phase precipitates are able to resist impact to energetic particle irradiation in a novel aluminium crossover alloy (<https://doi.org/10.1002/advs.202002397>). It has been reported that the T-phase is able to survive higher doses than other hardening phases in commercial aluminium alloys (i.e. Mg₂Si in 6xxx series Al-based alloys). One of the explanations provided by the authors to explain such a radiation tolerance is the fact that in aluminium crossover alloys, the volumetric fraction of T-phase precipitates is significantly higher than other phases in commercial aluminium alloys. If the number of precipitates divided by the volume is higher, then they act as a powerful sink for irradiation-induced defects. This reviewer question if the same thing happens here in this discovery reported by Y. Wang et al. in this present submission? Perhaps, the enhanced resistance to hydrogen embrittlement arises since upon precipitation of T-phase precipitates, the precipitation density increases considerably compared to eta-prime, leading to higher sinking efficiency for hydrogen.

Response: The reviewer pointed out an important issue about the volume fraction of T

phase. Existing studies indicated that T phase tends to form at higher aging temperature (>150 degC), whereas fully η phase was found in low-temperature (such as 120 degC) aged Al-Mg-Zn(-Cu) alloys. Please see references such as: Lee et al., Journal of Japan of Institute of Light Metals, 2017, <https://doi.org/10.2464/jilm.67.162>. That's why we decided to elevate the aging temperature from 120 to 150 degC. As shown in the SAED patterns in Fig. 1b and Fig. 1d, the characteristic spots for T phase appears in HT material. This means the **volume fraction of T in HT material has been successfully increased**, leading to higher sinking efficiency for hydrogen, as suggested by the reviewer.

Moreover, other than volume fraction (trap density) of T phase, it is believed that the **hydrogen trapping features** of T phase play even more important roles. On one hand, compared to the hydrogen trapping at 2D interfaces at $MgZn_2$, T phase strongly traps hydrogen in **its interior (3D space)**, which means single T phase particle is capable of trapping more hydrogen atoms; on the other hand, the **hydrogen trap energy** at T phase is much higher than that of coherent $MgZn_2$ interfaces, which endows T phase with strong hydrogen trapping capacity, because hydrogen trapping is strongly related to the binding energy at trap sites, as illustrated by the following **Fig. R8**:

Fig. R8 dependence of hydrogen occupancy at trap sites on their hydrogen trap energies

It is seen from **Fig. R8** that when the trap energy is elevated by 0.1 eV, hydrogen occupancy can be increased by around 2 orders of magnitude.

More discussions regarding these aspects are provided in the revised manuscript.

In conclusion, we have tried our best to improve the quality of this manuscript and all the comments from three reviewers have been addressed. We hope the revised version is acceptable for publication.

We would like to thank the reviewers again for your time and consideration.

Best wishes.

Yafei Wang; yfwangxjtu@outlook.com

REVIEWERS' COMMENTS

Reviewer #2 (Remarks to the Author):

All my prior questions have been answered. I have no more further questions.

Reviewer #3 (Remarks to the Author):

The manuscript has been significantly increased in both quality and clarity after this review session. All my comments have been satisfied by the authors who even performed additional experiments to prove their claims. I recommend the publication of the manuscript in this presented revised version.

Thank you.